# How generalized hydrodynamics time evolution arises from a form factor expansion

Axel Cortés Cubero [*]

Institute for Theoretical Physics, University of Amsterdam, Science Park 904, 1098 XH
Amsterdam, The Netherlands

## Abstract

The generalized hydrodynamics (GHD) formalism has become an invaluable tool for the study of spatially inhomogeneous quantum quenches in (1+1)-dimensional integrable models. The main paradigm of the GHD is that at late times local observables can be computed as generalized Gibbs ensemble averages with space-time dependent chemical potentials. It is, however, still unclear how this semiclassical GHD picture emerges out of the full quantum dynamics. We evaluate the quantum time evolution of local observables in spatially inhomogeneous quenches, based on the quench action method, where observables can be expressed in terms of a form factor expansion around a finite-entropy state. We show how the GHD formalism arises as the leading term in the form factor expansion, involving one particle-hole pair on top of the finite-entropy state. From this picture it is completely transparent how to compute quantum corrections to GHD, which arise from the higher terms in the form factor expansion. Our calculations are based on relativistic field theory results, though our arguments are likely generalizable to generic integrable models.

## 1 Introduction

Quantum integrable systems are an ideal laboratory for the study of non-equilibrium phenomena. While such systems can be strongly interacting, and display rich phenomenology, integrability still provides useful analytical control [1–3].

A recent breakthrough in the field of non-equilibrium quantum integrable systems was the introduction of the theory of generalized hydrodynamics (GHD), discovered almost simultaneously in [4,5]. GHD provides a relatively simple mathematical framework that allows for computations in spatially inhomogenous non-equilibrium conditions.

GHD was initially introduced as a framework to evolve a spatially inhomogeneous initial state with the Hamiltonian of a homogeneous integrable system. At late times, it is assumed that the space-time dependence of physical observables is weak enough, that observables may be computed in terms of a local generalized Gibbs ensemble (GGE). GHD then provides an evolution equation which relates the parameters of the GGE at different times and positions. This formalism has also been extended and generalized in several ways, and been applied to a wide range of spatially inhomogeneous problems [6,6–23]. The validity of GHD has also recently been tested in experiments [24].

GHD is a semiclassical effective description, which is expected to apply at large spatial and time scales. The GHD evolution equations are derived from a kinematical picture, by assuming that one only needs to study the motion of quasi-particle excitations and their effective scattering matrix. A consequence of this is that GHD (as originally introduced) can only describe ballistic transport. Within this same semiclassical framework, there have been attempts to perturb away from this limit, allowing for some quantum fluctuations which would account diffusive effects by including Navier-Stokes terms on the GHD evolution equations

---

[*]ax.cortescubero@gmail.com

[25,26]. An alternate approach in [27], treated the GHD as the classical equations of motion which minimize some action, then compute quantum corrections as fluctuations around this minimal action configuration.

Despite the undeniable success of GHD, it is yet to be understood exactly how this semiclassical evolution arises from purely quantum first principles. It is thus not understood what kind of approximation GHD actually is, in terms of the full quantum description, and thus it is hard to understand how to compute the leading quantum corrections.

In this paper we will apply recent results about the form factor expansion for correlation functions of integrable quantum field theories to understand how GHD emerges for some class of spatially inhomogeneous initial states. We will use the tools of the recently developed thermodynamic bootstrap program (TBP) [28, 29], which was introduced as an axiomatic approach to compute form factors of local operators with particle and hole excitations on top of some finite-entropy thermodynamic background state (in contrast to the standard bootstrap program which concerns particle excitations on top of the vacuum).

We will show that GHD time evolution corresponds in the form factor expansion, to including only up to certain one-particle-hole pairs form factors corresponding to a spatially inhomogeneous initial state. We will see explicitly using the results from the TBP from [29] that by keeping only this contribution, the full GHD description and evolution equations are recovered. It is then clear within our formalism what we need to do to compute corrections to the GHD limit: we need to include subleading corrections from other form factors with higher number of particles and holes.

We focus on the study of relativistic quantum field theories, purely because it is there where the useful results from the TBP apply, and we have explicit expressions for some form factors. It could be expected, however, that the arguments presented in this paper are more general than QFT, and a similar derivation can be done for integrable models of non-relativistic particles, and quantum spin chains. There have been also some recent developments in the computation of thermodynamic form factors in such models [30, 31], so there is hope that an analogous derivation as the one presented here can be done for all kinds of quantum integrable models.

The structure of the paper is as follows. In the next section we will introduce some necessary tools and notations from integrable QFT, and their thermodynamic description through the thermodynamic Bethe ansatz. In Section 3, we introduce the quench action method, which is a useful analytical tool introduced in [32], which facilitates the computation of expectation values of local operators after a quantum quench. In Section 4 we introduce a class of spatially inhomogenous initial states, and recall how GHD can be applied to such states to describe time-evolution at late times. We then show how inhomogeneity adds certain difficulties to the application of the quench action formalism. In Section 5 we present a brief introduction to the TBP formalism, and recall some useful results that will be needed in our derivation. In Section 6, as a warm up exercise, we compute the expectation values of local operators at $t = 0$ in the inhomogenous initial-states we have introduced, and show that these can be computed in terms of a local GGE, as is expected. In Section 7 we compute the leading term in the form factor expansion, for the expectation value of local observables at late times after the inhomogeneous initial state. We will show that the full GHD evolution is recovered from this leading contribution corresponding to one-particle-hole pair form factors. In Section 8 we show how next-to-leading corrections to local observables at late times can be computed by including higher form factor terms. We present our conclusions in Section 9.

# 2   Integrable QFT in the thermodynamic limit

In this section we will introduce several concepts from the thermodynamic Bethe ansatz that will be necessary in the following section. For most of this paper we will concentrate mostly on relativistic quantum field theories, for simplicity, even though we expect our results to be applicable to more generic integrable models.

We will concentrate on QFT's with one species of particle (such as sinh-Gordon), for simplicity of presentation, even though it is not too difficult to generalize our results to other *diagonal scattering* (where particle don't exchange species upon scattering) theories with several species of particles. In a relativistic

QFT, the particle energy and momentum can be parametrized in terms of a rapidity parameter, $\theta$, as

$$E = m \cosh\theta, \ \ p = m \sinh\theta, \tag{1}$$

respectively, where $m$ is the particle mass.

Integrable QFT's are characterized by elastic and factorizable scattering, where all scattering events can be factorized into a product of two-particle S-matrices. We denote the two-particle S-matrix as $S(\theta_{12})$, which depends only on the difference of rapidities of the two particles $\theta_{12} = \theta_1 - \theta_2$.

We will be interested in studying the thermodynamic limit, where we consider a QFT with system size $L$, and states with a number of particles $N$, and take both quantities to infinity, keeping a finite particle density, $L \to \infty$, $N \to \infty$, with $L \sim N$. Such thermodynamic states are then more conveniently parametrized by introducing a particle density function, $\rho_p(\theta)$, which gives the probabiity of finding a particle in the state with rapidity in a small interval $\theta + \Delta\theta$. In the thermodynamic limit it is then useful to label states in terms of the particle densities $|\rho_p\rangle$, instead of the full set of particle rapidities. We will assume throughout this paper that these states have been normalized as $\langle \rho_p | \rho_p \rangle = 1$.

It is also useful to introduce the *density of states*, $\rho_s$, which according to standard thermodynamic Bethe ansatz (TBA) [33] calculations is related to the density of particles by

$$\rho_s(\theta) = m \cosh(\theta) + \int d\theta' T(\theta - \theta')\rho_p(\theta'), \tag{2}$$

where we define

$$T(\theta) \equiv \frac{1}{2\pi}\frac{\partial}{\partial\theta}\delta(\theta), \quad \delta(\theta) \equiv -i\log(-S(\theta)). \tag{3}$$

An important property of such states, $|\rho_p\rangle$, is that following the TBA formalism, when one introduces an additional particle on top of this thermodynamic state, the rapidities of all the other particles in the background are shifted by an amount of order $1/L$, but given that there are $N \sim L$ background particles, the total shift to physical quantities is finite.

In particular, we can consider conserved charges, $Q_i$, which have one-particle eigenvalues given by

$$Q_i|\theta\rangle = h_i(\theta)|\theta\rangle. \tag{4}$$

For instance these could be the energy or momentum, with $h(\theta) = m \cosh\theta, m \sinh\theta$, respectively. When we consider a particle excitation of rapidity $\theta$, created on top of the thermodynamic state $|\rho_p\rangle$, then we can define a "dressed" charge as

$$h_i^{\rm dr}(\theta) = h_i(\theta) + \int d\theta T(\theta' - \theta')n(\theta')h_i^{\rm dr}(\theta'), \tag{5}$$

where we have introduced the filling fraction, defined as $n(\theta) = \rho_p(\theta)/\rho_s(\theta)$. This dressed quantity is related to the effective charge of the particle (the charge of the particle itself, plus the total change in the charge of the background particles),

$$\frac{\langle \rho_p; \theta | Q_i | \rho_p; \theta \rangle}{\langle \rho_p; \theta | \rho_p; \theta \rangle} - \frac{\langle \rho_p | Q_i | \rho_p \rangle}{\langle \rho_p | \rho_p \rangle} = h_i^{\rm eff}(\theta), \tag{6}$$

as,

$$h_{\rm eff}'(\theta) = (h')^{\rm dr}(\theta). \tag{7}$$

One particularly important quantity for our purposes will be the *effective velocity* of a particle

$$v^{\rm eff}(\theta) = \frac{(E')^{\rm dr}(\theta)}{(p')^{\rm dr}(\theta)}, \tag{8}$$

which describes how fast a particle of rapidity $\theta$ is able to move in the presence of a thermodynamic background described by $\rho_p(\theta)$.

# 3 Quench action for pure states and density matrices

In this section we briefly review the quench action formalism for spatially homogeneous quantum quenches of integrable models. We will largely follow the discussion presented in the review paper [34] (see also references within the review for different problems where this method has been applied). The quench action method was originally formulated to work with initial conditions described by a pure state, but here we will show how this can be generalized to the case of a generic density matrix.

A quantum quench consists on initializing a system with Hamiltonian, $H$, in a state, $|\Psi_0\rangle$, which is not an eigenstate. The state can be expressed in terms of the basis of eigenstates,

$$|\Psi_0\rangle = \sum_n c_n|n\rangle, \tag{9}$$

where we define the overlaps $c_n = \langle n|\Psi_0\rangle$, and the eigenstates $H|n\rangle = E_n|n\rangle$. One is then interested in computing time-dependent expectation values of local operators,

$$\langle \mathcal{O}(x,t)\rangle = \frac{\langle \Psi_t|\mathcal{O}(x,0)|\Psi_t\rangle}{\langle \Psi_0|\Psi_0\rangle}, \tag{10}$$

where $|\Psi_t\rangle = \sum_n e^{-iE_n t} c_n|n\rangle$. This expectation value can be computed in principle, if one knows all the overlaps, $c_n$, and all the matrix elements of the operator evaluated on eigenstates of the Hamiltonian,

$$\langle \mathcal{O}(x,t)\rangle = \frac{\sum_{n,m} c_n c_m^* e^{-it(E_n - E_m)} \langle m|\mathcal{O}(x,0)|n\rangle}{\sum_n |c_n|^2}. \tag{11}$$

The necessary ingredients to compute time-dependent expectation values after a quantum quench are then knowledge of the complete basis of states, $|n\rangle$, knowledge of the overlaps between the initial state and this basis, and knowledge of matrix elements of operators in this basis.

Integrable theories are characterized by elastic and factorizable scattering of particle excitations. The eigenstates can be labeled as $|n\rangle = |\theta_1, \ldots, \theta_n\rangle$, understood as multi-particle states, with a rapidity parameter for each particle. Elasticity means that the set of particle rapidities $\{\theta\}$ are conserved quantities.

The next necessary ingredient is to know matrix elements of local operators in multi-particle states, of the form $\langle \{\theta'\}|\mathcal{O}(x,t)|\{\theta\}\rangle$, where $\{\theta\}$ denotes a given set of rapidities. Such matrix elements can be obtained in the case of relativistic field theories using a bootstrap approach [35]. We will review further the structure of the necessary matrix elements in the next sections.

Lastly, for some quantum quenches, it is possible to obtain all the overlaps, of the form $\langle \{\theta\}|\Psi_0\rangle$. One useful approach, for example, to obtain solvable initial states (those where all overlaps can be computed) has been to consider states which correspond to integrable boundary conditions in the crossed channel [36, 37], where the role of space and time are exchanged. Obtaining overlaps for general quench protocols, though, is a hard problem and an ongoing subject of investigation [38–45]

Even when one knows all the necessary ingredients to compute the time evolution of local observables, it is still a difficult problem to extract meaningful information out of the expansion (11). This is because this expression involves a double sum over the entire Hilbert space, and it is not clear initally which terms in this expansion are the most important that one needs to keep, or how the expansion may be in any way resummed.

The quench action approach [32, 34] was introduced as a way to alleviate this difficulty. This approach states that for a system in the thermodynamic limit, at any finite time, the double sum (11) can be replaced with a single sum over the Hilbert space. Furthermore, the expectation value at infinite time is given by evaluating a single matrix element on a "representative state".

Global quenches generally introduce an extensive amount of energy into the system. The initial state will then have large overlaps with many-particle states $|\theta_1, \ldots, \theta_n\rangle$, where $n \sim L$, and $L$ is the system size. In the thermodynamic limit, one considers then states with an infinite number of particles. States are then more conveniently parametrized in terms of the particle density function, $\rho_p(\theta)$.

For each state $|\rho_p\rangle$, there is a large number of microscopic configurations of particles $\{\theta\}$, which lead to the same distribution, $\rho_p(\theta)$ in the thermodynamic limit. For instance, taking the state $|\rho_p(\theta)\rangle$ and adding or removing a finite set of particles, will not change the macroscopic distribution. The number of microscopic states which lead to the same distribution $\rho_p(\theta)$ is given by $\exp(-S_{YY}[\rho_p])$, where $S_{YY}[\rho_p] \sim L$ is known as the Yang-Yang entropy.

Let's first consider the denominator of (11), $D = \sum_n |c_n|^2$, in the thermodynamic limit, we can replace the sum over $n$ with a path integral over distributions,

$$D = \int \mathcal{D}\rho_p \, \exp{-S_{QA}[\rho_p]}, \tag{12}$$

where we have defined $\exp\{-S_{QA}[\rho_p]\} = \exp\{-2\mathbb{RE}\{S_\Psi[\rho_p]\} - S_{YY}[\rho_p]\}$, and $\exp\{-S_\Psi[\rho_p]\} \equiv \langle\Psi_0|\rho_p\rangle$. We call the quantity $S_{QA}[\rho_p]$ the "Quench action", as the expression (12) is reminiscent of the path integral of some field theory.

A key insight of the quench action approach is noticing that $S_{QA}[\rho_p] \sim L$, therefore in the thermodynamic limit, the action is very large, such that the saddle point approximation of the path integral becomes exact,

$$\lim_{L\to\infty} D = e^{-S_{QA}[\rho_p^{\mathrm{sp}}]}, \tag{13}$$

where we define the saddle point configuration as $\delta S_{QA}[\rho_p]/\delta\rho_p|_{\rho_p=\rho_p^{\mathrm{sp}}} = 0$. The great advantage of this approach is that in the thermodynamic limit, the infinite sum over states, $\sum_n$, has been replaced by considering only one representative eigenstate, given by $|\rho_p^{\mathrm{sp}}\rangle$.

A similar logic can be applied to the numerator, including a local operator,

$$\mathcal{N} = \sum_{n,m} c_n c_m^* e^{-it(E_n-E_m)} \langle m|\mathcal{O}(x,0)|n\rangle. \tag{14}$$

This expression is more complicated than the denominator we have just evaluated, because it contains a double sum over states, yet it is easy to see that the quench action method reduces this to a single sum of states centered around saddle point, as long as the operator satisfies certain conditions.

We start the evaluation by replacing the sum over $m$, in the thermodynamic limit with a path integral over distributions $\rho_p$,

$$\mathcal{N} = \frac{1}{2} \int \mathcal{D}\rho_p \sum_n c_n \, , e^{-S_\Psi[\rho_p]-S_{YY}[\rho_p]-\mathrm{i}\Delta E_n[\rho_p]t+\mathrm{i}\Delta P_n[\rho_p]x} \langle\rho_p|\mathcal{O}(0,0)|n\rangle + \mathrm{C.C.}, \tag{15}$$

where $\Delta E_n[\rho_p]$ and $\Delta P_n[\rho_p]$ are the differences in total energy and momentum, respectively, between the states $|n\rangle$ and $|\rho_p\rangle$. The C.C. corresponds to the possibility of having started by replacing the sum over $n$ first with a path integral, instead of replacing first the sum over $m$.

There are simplifications that arise when we consider $\mathcal{O}$ to be a local operator (a more precise definition of the conditions that the operator needs to satisfy can be found in [34]). In this case, we can assume that when the operator acts locally on a highly-energetic state, it will not modify the state by a macroscopically large amount. More precisely, in the thermodynamic limit, the matrix elements, $\langle\rho_p|\mathcal{O}(0,0)|n\rangle$ are only non-zero if $|n\rangle = |\rho_p; \{\theta\}\rangle$, where $\{\theta\}$ is a finite set of modifications to the representative state, where the the total energy of the state is not modified by a macroscopic amount,

$$\langle\rho_p|H|\rho_p\rangle = \langle\rho_p; \{\theta\}|H|\rho_p\{\theta\}\rangle \left(1 + \mathcal{O}\left(\frac{1}{L}\right)\right). \tag{16}$$

As the state $|n\rangle$ is thermodynamically close to the representative state, we assume also the overlap of the state with the initial state can be written as $c_n = \exp\{-S_\Psi^*[\rho_p] - \delta S_\Psi[\rho_p, \{\theta\}]\}$, where $\delta S_\Psi[\rho_p, \{\theta\}] \sim L^0$ is a thermodynamically intensive quantity. This means that when we compute the saddle point of the quench action, it will not be modified by this small perturbation to the quench action.

We can now write

$$\mathcal{N} = \int \mathcal{D}\rho_p \sum_{\{\theta\}} e^{-S_{QA}[\rho_p] - \delta S_\Psi[\rho_p, \{\theta\}] - i\varepsilon_{\rho_p}(\{\theta\})t + ik_{\rho_p}(\{\theta\})x} \langle \rho_p | \mathcal{O}(0,0) | \rho_p; \{\theta\}\rangle, \tag{17}$$

where $\varepsilon_{\rho_p}(\{\theta\})$ and $k_{\rho_p}(\{\theta\})$ can be interpreted as the effective energy and momentum, respectively, of the set of excitations, $\{\theta\}$ on top of the representative state. In this form, it is now evident that for local operators, the saddle point of the quench action is not modified by the additional excitations $\{\theta\}$, therefore in the thermodynamic limit we can write the numerator as

$$\mathcal{N} = \frac{1}{2} \sum_{\{\theta\}} e^{-S_{QA}[\rho_p^{\mathrm{sp}}] - \delta S_\Psi[\rho_p^{\mathrm{sp}}, \{\theta\}] - i\varepsilon_{\rho_p^{\mathrm{sp}}}(\{\theta\})t + ik_{\rho_p^{\mathrm{sp}}}(\{\theta\})x} \langle \rho_p^{\mathrm{sp}} | \mathcal{O}(0,0) | \rho_p^{\mathrm{sp}}; \{\theta\}\rangle + \mathrm{C.C.} \tag{18}$$

The expectation value of the local operator can now be written as

$$\langle \mathcal{O}(x,t)\rangle = \frac{\mathcal{N}}{D} = \frac{1}{2} \sum_{\{\theta\}} e^{-\delta S_\Psi[\rho_p^{\mathrm{sp}}, \{\theta\}] - i\varepsilon_{\rho_p^{\mathrm{sp}}}(\{\theta\})t + ik_{\rho_p^{\mathrm{sp}}}(\{\theta\})x} \langle \rho_p^{\mathrm{sp}} | \mathcal{O}(0,0) | \rho_p^{\mathrm{sp}}; \{\theta\}\rangle + \mathrm{C.C.}, \tag{19}$$

where, as anticipated, the double sum over $n, m$ has been replaced by the single sum over excitations $\{\theta\}$ around the representative state.

For a spatially homogeneous initial state, the local observables are $x$-independent, as the initial state is annihilated by the total momentum operator. In this case it is also easy to see that at large times, contributions from excitations which produce large energy differences, $\varepsilon_{\rho_p^{\mathrm{sp}}}(\{\theta\})$ become highly oscillatory, and dephase as we integrate over all combinations $\{\theta\}$. Therefore in the infinite time limit we expect equilibration to the stationary value

$$\lim_{t \to \infty} \langle \mathcal{O}(x,t)\rangle = \langle \rho_p^{\mathrm{sp}} | \mathcal{O}(x,0) | \rho_p^{\mathrm{sp}}\rangle, \tag{20}$$

such that they become time-independent, and the system locally equilibrates.

It is important at this point to remark that the applicability of the saddle point approximation, leading to the late time equilibration relies on the assumption that the quench action has a single saddle point. Furthermore, it is assumed that the action is steep enough, (and that it does not broaden as we increase $L$), such that the path integral localizes to only the saddle point contribution. As we will see later, these assumptions will be challenged in the case where we have spatially inhomogeneous initial conditions.

For our purposes, we will need to generalize the quench action method to the case where the initial conditions are described by a non-equilibrium density matrix, rather than a pure state. As far as we know, this generalization, although fairly straightforward, has not been previously studied. In this case we write the time dependent expectation values of local observables as

$$\langle \mathcal{O}(x,t)\rangle = \frac{\mathrm{Tr}\left(\varrho_t \mathcal{O}(x,0)\right)}{\mathrm{Tr}(\varrho_0)}, \tag{21}$$

where $\varrho_t = e^{-iHt} \varrho_0 e^{iHt}$, where non-equilibrium initial conditions are characterized by the fact that $\varrho_0$ is not diagonal in the basis of eigenstates of the Hamiltonian.

We can proceed similarly by studying first the denominator, which is a simpler quantity, where we express the trace explicitly as a sum over eigenstates

$$D = \mathrm{Tr}(\varrho_0) = \sum_n \langle n | \varrho_0 | n\rangle. \tag{22}$$

Taking the thermodynamic limit, we can again replace the sum over states with a path integral over distributions,

$$D = \int \mathcal{D}\rho_p e^{-S_{QA}[\rho_p]}, \tag{23}$$

where

$$e^{-S_{QA}[\rho_p]} = e^{-2S_\varrho[\rho_p]-S_{YY}[\rho_p]} = \langle\rho_p|\varrho_0|\rho_p\rangle e^{-S_{YY}[\rho_p]}. \tag{24}$$

Now similarly to what we did before, in the thermodynamic limit, the partition function localizes to only the contribution from the saddle point of the quench action. The only difference between the general density matrix, and the pure state case, is that the term $2S_\varrho[\rho_p]$ cannot be separated into a contribution coming from the bra and one from the ket, as we did in the pure state case. Nevertheless this separation is not necessary to build the quench action and find its saddle point.

Similarly we consider the numerator, again replacing the trace over states with a path integral over distributions

$$\mathcal{N} = \int \mathcal{D}\rho_p e^{-S_{YY}[\rho_p]} \langle\rho_p|\varrho_t\mathcal{O}(x,0)|\rho_p\rangle. \tag{25}$$

We can now insert a complete set of states between the local operator and the density matrix

$$\mathcal{N} = \int \mathcal{D}\rho_p \sum_n e^{-S_{YY}[\rho_p]} \langle\rho_p|\varrho_t|n\rangle\langle n|\mathcal{O}(x,0)|\rho_p\rangle. \tag{26}$$

Again we can assume that if the operator is local, then the numerator will only have nonzero contributions if $|n\rangle = |\rho_p, \{\theta\}\rangle$, with only a thermodynamically intensive number of excitations around the representative state. We can also assume, as we did before, that the projection of the density matrix on these excited states, are thermodynamically close to those on the representative state,

$$\langle\rho_p|\varrho_0|\rho_p, \{\theta\}\rangle = e^{-2S_\varrho[\rho_p]-\delta S_\varrho[\rho_p,\{\theta\}]}, \tag{27}$$

with $\delta S_\varrho[\rho_p, \{\theta\}] \sim L^0$.

Having thus defined what is meant by the quench action corresponding to a initial density matrix, the rest of the computation from this point onwards follows exactly as the pure state case, such that we do not need to repeat the same derivation. The same result (19) applies for the time-dependent expectation values, with the only modification that the quench action is defined by Eq. (27).

In the next section we will introduce a certain class of spatially-inhomogeneous initial states. We will see how the assumptions we made about the saddle point of the quench action generally break down in the inhomogeneous case.

## 4   Inhomogeneous initial states

### 4.1   GHD approach to inhomogeneous quenches

Quantum integrable QFT's are characterized by the presence of an extensive number of conserved charges, $[Q_i, H] = 0$, that are said to be local, in the sense that they can be expressed as the spatial integral over some charge density operator,

$$Q_i = \int dx q_i(x). \tag{28}$$

A generalized Gibbs ensembe (GGE) can be constructed by specifying a different chemical potential $\beta^i$ corresponding to each of the conserved charges, then averages of local observables may be computed as

$$\langle\mathcal{O}\rangle_{GGE} = \frac{\text{Tr}\left(e^{-\sum_i \beta^i Q_i}\mathcal{O}\right)}{\text{Tr}\, e^{-\sum_i \beta^i Q_i}}. \tag{29}$$

We emphasize that observables computed in a GGE are time-independent, since all the charges commute with the Hamiltonian, such that $\varrho_t = e^{-\mathrm{i}Ht}\varrho_0 e^{\mathrm{i}Ht} = \varrho_0$, for $\varrho_0 = \exp\left(\sum_i \beta^i Q_i\right)$.

In the thermodynamic limit, local observables in the GGE may also be computed using a representative state approach, as discussed in the previous section. The ensemble average may be replaced by averaging on a single representative state of the Hamiltonian $|\rho_p^{GGE}\rangle$. The rapidity distribution $\rho_p^{GGE}(\theta)$ can be computed as the saddle point of the quench action

$$S_{GGE}[\rho_p] = \langle\rho_p| \sum_i \beta^i Q_i |\rho_p\rangle + S_{YY}[\rho_p]. \tag{30}$$

Given that all the charges commute with the Hamiltonian, the state $|\rho_p\rangle$ is also an eigenstate of each of the charges, and the eigenvalues can be generally written as a linear functional of the distribution,

$$Q_i|\rho_p\rangle = L \int d\theta h_i(\theta)\rho_p(\theta)|\rho_p\rangle, \tag{31}$$

with a particular function $h_i(\theta)$ specified for each different charge, defined in (4). We point out that the eigenvalues of the conserved charges in the representative are extensive, they grow linearly with system size. The conserved charges thus contribute only linear terms in the $\rho_p(\theta)$ to the quench action. Such a linear quench action has already been studied, in [46], where it arises describing a quantum quench in the sinh-Gordon model. The corresponding saddle point equation is derived in [46] and solved numerically, where good convergence to one saddle point is shown. The functional derivative of this linear term of the quench action does not depend on $\rho_p(\theta)$, so this adds a simple driving term to the saddle point equation, $\delta S_{GGE}/\delta\rho_p$.

We will now consider a class of initial states based on a modification of the GGE, which was recently extensively studied in [8]. Our initial conditions are given by a generalization of the GGE, where chemical potentials are also position dependent. Our observables are then given by (at $t = 0$),

$$\langle\mathcal{O}(x,0)\rangle_{LGGE} = \frac{\text{Tr}\left(e^{-\int dy \sum_i \beta^i(y)q_i(y)} \mathcal{O}(x,0)\right)}{\text{Tr}e^{-\int dy \sum_i \beta^i(y)q_i(y)}}, \tag{32}$$

where the initial state is fixed by specifying the set of functions $\beta^i(y)$, and the subscript on the left-hand-side stands for "local GGE".

For our purposes, we will further demand that the functions $\beta^i(y)$ are piece-wise very smooth. That is, we demand that $d\beta^i(y)/dy$ is very small everywhere, except for a finite number of points, where the function is allowed to suddenly jump. We will later specify what we mean by "very small" here. We allow for sudden jumps in the chemical potentials to accommodate common protocols, such as the bi-partition studied in [4,5], where $\beta^i(y) = \delta^{i0}(\beta + \Theta(y)\beta')$, where we have identified $Q_0 = H$.

In [8] it was postulated that given such an initial state, expectation values of operators at $t = 0$, may be computed through a local density approximation, i.e. as a standard GGE average, where the value of $x$ is only important for specifying the repesentative state. That is, it is assumed that for piece-wise very smooth initial states, $t = 0$ expectation values may be evaluated as

$$\langle\mathcal{O}(x,0)\rangle_{LGGE} = \langle\rho_p^{GGE}(x)|\mathcal{O}|\rho_p^{GGE}(x)\rangle, \tag{33}$$

where $|\rho_p^{GGE}(x)\rangle$ is a spatially homogeneous eigenstate, specified by the distribution $\rho_p^{GGE}(x;\theta)$, which is the saddle point of the quench action

$$S_{GGE}(x)[\rho_p] = \sum_i \beta^i(x)\langle\rho_p|Q_i|\rho_p\rangle + S_{YY}[\rho_p]. \tag{34}$$

From our point of view this statement is not self evident, and this is something that we need to show, which we will do in the following sections.

The main assumption of generalized hydrodynamics is that after a spatially inhomogeneous quench, at long enough times and distances, expectation values of local observables can be computed as expectation

values of a GGE, where the space and time dependence only comes into the determination of the particle density distribution, $\rho_p^{GGE}(x,t;\theta)$. Expectation values can then by expressed as

$$\langle \mathcal{O}(x,t)\rangle_{LGGE} = \langle \rho_p^{GGE}(x,t)|\mathcal{O}|\rho_p^{GGE}(x,t)\rangle. \tag{35}$$

Under this assumption, it is possible to derive a simple differential equation that governs the dynamics of the distribution $\rho_p^{GGE}(x,t;\theta)$. It is more convenient to work in terms of the *filling fraction*, $n(x,t;\theta) \equiv \rho_p^{GGE}(x,t;\theta)/\rho_s^{GGE}(x,t;\theta)$, where $\rho_s^{GGE}(x,t;\theta)$ is the density of states. The GHD evolution equation is then [4, 5],

$$\partial_t n(x,t;\theta) + v^{\mathrm{eff}}(x,t;\theta)\partial_x n(x,t;\theta) = 0, \tag{36}$$

where $v^{\mathrm{eff}}(x,t;\theta)$ is the effective velocity of a particle of rapidity $\theta$ travelling in the GGE background described by the filling fraction $n(x,t;\theta)$.

The differential equation (36) can be solved using as initial conditions the filling fractions $n(x,0;\theta)$ given by the saddle points of the ($x$-dependent) quench action (30). The solution can be expressed as [9],

$$n(x,t;\theta) = n(u(x,t;\theta),0;\theta), \tag{37}$$

where,

$$\int_{x_0}^{x} dy\rho_s(y,t;\theta) = \int_{x_0}^{u(x,t;\theta)} dy\rho_s(y,0;\theta) + v^{\mathrm{eff}}(x_0,0;\theta)\rho_s(x_0,0;\theta)t, \tag{38}$$

where $x_0$ is a negative number chosen to be large enough such that $n(x,f,\theta) = n(x,0,\theta)$, for all $x < x_0$ and $f \in [0,t]$.

The solution (37) has a simple interpretation. The point $u(x,t;\theta)$ can be interpreted as the spatial position at time 0, from which a particle with rapidity $\theta$ would reach a position $x$ at time $t$, traveling with the effective velocity.

## 4.2 Quench action approach and multiplicity of saddle points

In the remainder of this section, we will explore what happens if we attempt to apply the same quench action logic we discussed in the previous section, but when we consider initial states described by the spatially inhomogeneous density matrix, $\varrho_0 = \exp\left(-\int dy \sum_i \beta^i(y)q_i(y)\right)$. We begin by considering the denominator $D = \sum_n \langle n|\varrho_0|n\rangle$. Again, in the thermodynamic limit, we may replace the sum over states by a path integral over distributions,

$$D = \int \mathcal{D}\rho_p\, e^{-S_{YY}[\rho_p]}\langle\rho_p|e^{-\int dy \sum_i \beta^i(y)q_i(y)}|\rho_p\rangle, \tag{39}$$

The matrix element in (39) can be computed using the expansion,

$$\langle\rho_p|e^{-A}|\rho_p\rangle = e^{-\langle\rho_p|A|\rho_p\rangle + \frac{1}{2}\left(\langle\rho_p|A^2|\rho_p\rangle - (\langle\rho_p|A|\rho_p\rangle)^2\right) + \dots} \tag{40}$$

in terms of the *connected* parts of the expectation values, $\langle\rho_p|A^n|\rho_p\rangle_{\mathrm{connected}}$. We notice that in the spatially homogeneous limit, where $\beta^i(y) = \beta^i$, all the higher order terms vanish, since there is not connected part, and $\langle\rho_p|\left(\sum_i \beta^i Q_i\right)^n|\rho_p\rangle = \left(\langle\rho_p|\sum_i \beta^i Q_i|\rho_p\rangle\right)^n$. The expansion (40) is therefore based on the magnitude of the inhomogeneity of the chemical potentials, $\beta^i(y)$. For slowly varying chemical potentials, the higher order terms become small corrections.

We examine the second order term in (40),

$$S_{QA}^{(2)}[\rho_p] \equiv \langle\rho_p|\left(\int dy \sum_i \beta^i(y)q_i(y)\right)^2|\rho_p\rangle - \left(\langle\rho_p|\int dy \sum_i \beta^i(y)q_i(y)|\rho_p\rangle\right)^2. \tag{41}$$

As our notation suggests, this term can be interpreted as a contribution to the quench action from the quadratic connected term. For the purposes of our discussion, we do not need to evaluate this term explicitly (even though it can be done, since expectation values of charge densities can be computed). We only point out that when the this term of the quench action is a quadratic functional of the particle distribution,

$$S_{QA}^{(2)}[\rho_p] = \int d\theta_1 d\theta_2 \, h(\theta_1, \theta_2)\rho_p(\theta_1)\rho_p(\theta_2). \tag{42}$$

for some function $h(\theta_1, \theta_2)$. When we compute the functional derivative $\delta S_{QA}^{(2)}/\delta \rho_p$ this will add a term which is linear in $\rho_p(\theta)$ to the saddle point equation

This statement can be easily generalized, the $n$-th connected term of the quench action contains the expectation value of the product of $n$ charge densities, this leads generally to a contribution to the quench action of the form

$$S_{QA}^{(n)}[\rho_p] = \int d\theta_1, \ldots, d\theta_n \, h(\theta_1, \ldots, \theta_n)\rho_p(\theta_1) \ldots \rho_p(\theta_n). \tag{43}$$

The $n$-th term will thus add a term which is of order $n-1$ in $\rho_p(\theta)$ to the saddle point equation. In this way when the initial state is inhomogeneous, a polynomial potential is added to the saddle point equation. Even if the simple saddle point equation for the homogenous case had only one saddle point, in general, depending on the nature of the functions $h(\theta_1, \ldots, \theta_n)$, the polynomial potential will have numerous minima, which may lead to the existence of many possible saddle points.

The full quench action corresponding to to the inhomogeneous initial state (32) can then be written as the infinite expansion

$$S_{QA}[\rho_p] = \left(\sum_n S_{QA}^{(n)}[\rho_p]\right) + S_{YY}[\rho_p], \tag{44}$$

which may in general have an infinite number of saddle points, as the saddle-point equation has an infinite order polynomial potential in $\rho_p(\theta)$.

## 5   Review of thermodynamic form factors

We will briefly review in this section the recent development of thermodynamic form factors for quantum field theory. The Thermodynamic bootstrap program (TBP) was developed in [28] as an axiomatic formalism to compute matrix elements in integrable QFT of the form

$$f_{\rho_p}^{\mathcal{O}}(\theta_1, \ldots, \theta_n) = \frac{\langle \rho_p | \mathcal{O} | \rho_p; \theta_1, \ldots, \theta_n \rangle}{\langle \rho_p | \rho_p \rangle}, \tag{45}$$

that is, form factors concerning a finite number of particle excitations on top of a representative state characterized by the particle density distribution $\rho_p(\theta)$. It is important to remark that particles can not only be added to the thermodynamic state, but they can also be removed, equivalently one can introduce a "hole" with rapidity $\theta$. By Lorentz invariance, we know that introducing a hole with rapidity $\theta$ in the form factor is equivalent to introducing a particle with rapidity $\theta + \pi i$.

These form factors become relevant when we want to compute correlation functions of local operators on top of a thermodynamic state, $|\rho_p\rangle$. Two point functions can be written as the spectral decomposition,

$$\frac{\langle \rho_p | \mathcal{O}_1(x,t)\mathcal{O}_2(0,0) | \rho_p \rangle}{\langle \rho_p | \rho_p \rangle} = \sum_{n=0}^{\infty} \sum_{\sigma_i = \pm 1} \left( \prod_{k=1}^{n} \fint_{-\infty}^{\infty} \frac{d\theta}{2\pi} n_{\sigma_k}(\theta_k) \right) f_{\rho_p}^{\mathcal{O}_1}(\theta_1, \ldots, \theta_n)_{\sigma_1, \ldots, \sigma_n} \left( f_{\rho_p}^{\mathcal{O}_2}(\theta_1, \ldots, \theta_n)_{\sigma_1, \ldots, \sigma_n} \right)^*$$

$$\times \exp\left( i x \sum_{k=1}^{n} \sigma_k k(\theta_k) - i t \sum_{k=1}^{n} \sigma_k \omega(\theta_k) \right), \tag{46}$$

where we defined the label $\sigma_i = \pm 1$ do denote whether the excitation is a particle $(+)$ or a hole $(-)$, and,

$$n_{-1}(\theta) \equiv n(\theta), \quad n_{+1}(\theta) = \frac{\rho_s(\theta) - \rho_p(\theta)}{\rho_p(\theta)} n(\theta). \tag{47}$$

The integral sign $\fint$ denotes a particular regularization prescription that is defined in [28], as the form factors in the integrand generally feature poles in the real axis of rapidities.

It was proposed in [28] that these form factors on top of the thermodynamic background may be computed in an axiomatic, self consistent manner. This approach was called the thermodynamic bootstrap program(TBP). The set of axioms and how they can be used to compute form factors can be found in full detail in [28].

One main result that will be useful to us is the zero-momentum limit of the one-particle-hole pair form factor, which was derived in [29], which is given by

$$\lim_{\kappa \to 0} f_{\rho_p}^{\mathcal{O}}(\theta + \pi i, \theta + \kappa) = \sum_{k=0}^{\infty} \frac{1}{k!} \int \prod_{j=1}^{k} \left( \frac{d\theta_j}{2\pi} n(\theta_j) \right) f_c^{\mathcal{O}}(\theta_1, \ldots, \theta_k, \theta), \tag{48}$$

where $f_c^{\mathcal{O}}(\theta_1, \ldots, \theta_k, \theta)$ is the *connected* form factor, defined as the finite part of the form factor (without a thermodynamic background, $\rho_p(\theta) = 0$),

$$f_c^{\mathcal{O}}(\theta_1, \ldots, \theta_k, \theta) = \mathbf{F.P.} \lim_{\{\kappa_i\} \to 0} f^{\mathcal{O}}(\theta_1 + \pi i, \ldots, \theta_k + \pi i, \theta + \pi i, \theta_1 + \kappa_1, \ldots, \theta_k + \kappa_k, \theta + \kappa_{k+1}), \tag{49}$$

where the $\mathbf{F.P.}$ stands for the the finite part, defined as the term that remains finite when any of the $\kappa_i$ is taken individually to zero. The divergent properties of these form factors, as well as the proper regularization are thoroughly discussed in Refs. [47,48]. We point out also that this expression has been shown to simplify [8] in the case where the operator is a conserved charge density, for which it is found that Eq. (48) reduces to,

$$\lim_{\kappa \to 0} f^{q_i}(\theta + \pi i, \theta + \kappa) = h_i^{\mathrm{dr}}(\theta). \tag{50}$$

It has been shown that [29] the form factor (48) is enough to compute the Euler scale two-point functions of local operators. The one-particle (or one-hole) contributions to the two-point correlation functon are oscillatory functions of $x$, and $t$. Eulerian correlation functions are typically defined by performing an in-fluid-cell average in a local region around the point $x, t$, [8]. Contributions to the correlation function containing a different number of particles and antiparticles, being oscillatory, vanish after performing the in-cell average. Considering the asymptotic expansion of correlation function (46) at $t \to \infty$ with fixed $x/t = \xi$, and $\xi \in (-1, 1)$ (that is, for two causally connected operators, within each other's light-cone) the leading contribution (after in-cell averaging) comes from the one-particle-hole pair form factor (48). This can be understood by the fact that the $x, t$ dependent exponential factor, $\exp\left( i x \sum_{k=1}^{n} \sigma_k k(\theta_k) - i t \sum_{k=1}^{n} \sigma_k \omega(\theta_k) \right)$ becomes highly oscillatory at large times, and one can evaluate the asymptotic behavior of each term through a stationary phase approximation, yielding

$$\left[ \frac{\langle \rho_p | \mathcal{O}_1(\xi t, t) \mathcal{O}_2(0, 0) | \rho_p \rangle}{\langle \rho_p | \rho_p \rangle} - \frac{\langle \rho_p | \mathcal{O}_1 | \rho_p \rangle}{\langle \rho_p | \rho_p \rangle} \frac{\langle \rho_p | \mathcal{O}_2 | \rho_p \rangle}{\langle \rho_p | \rho_p \rangle} \right]^{\mathrm{Eulerian}}$$
$$= \frac{1}{t} \lim_{\kappa_1, \kappa_2 \to 0} \sum_{\theta \in \theta_*(\xi)} \frac{n(\theta)(1 - n(\theta))}{4\pi^2 \rho_s(\theta) |(v^{\mathrm{eff}})'(\theta)|} f_{\rho_p}^{\mathcal{O}_1}(\theta + \pi i, \theta + \kappa_1) f_{\rho_p}^{\mathcal{O}_2}(\theta + \pi i, \theta + \kappa_2)$$
$$+ \mathcal{O}\left( \frac{1}{t^2} \right), \tag{51}$$

whereas correlation functions outside of the lightcone, (with $|\xi| > 1$) are expected to decay exponentially, and $[\mathcal{O}_1(\xi t, t), \mathcal{O}_2(0, 0)] = 0$ for $|\xi| > 1$.

Corrections to the Euler scale correlator (51) come at higher orders of $1/t$, and arise from considering contributions from form factors with a higher number of particles and holes, as well as from corrections to the stationary phase approximation from the one-particle-hole pair form factors.

We point out that there are many aspects of the TBP formalism that are not tested or needed in the problem considered in this paper. The only aspect of the TBP we presently need is the assertion that in fact the two-point correlation function can be expressed in terms of a sum over dressed form factors, as defined in [28]. The TBP includes a set of axioms that can be used in the calculation of the form factors themselves, which we are not using here, since we will only need to use the one-particle-hole form factor in the zero-momentum limit (48), a calculation which was done in [29], based on the definition of the form factor, rather than computed directly from the axioms. If in the future we want to compute corrections to GHD dynamics, as discussed in Section 8, then we would need form factors outside of the zero-momentum limit, in which case we need to use and put to the test more of the TBP axioms.

## 6   Expectation values of local operators at $t = 0$

In this section we will show from quantum first principles how for an piecewise very smooth spatially inhomogeneous initial state, as defined in (32), the expectation values of local operators at $t = 0$, are given by their average on a locally defined GGE, as in (33). Even though this fact is treated as a starting assumption in [8], we find here it is useful to fully work out how this result arises within our coarse-grained approach, as once we have understood this, it is easier to study the expectation values at large $t$. The quantity we want to study is the expectation value,

$$\langle \mathcal{O}(x,0) \rangle_{LGGE} = \frac{\text{Tr}\left(e^{-\int dy \sum_i \beta^i(y) q_i(y)} \mathcal{O}(x,0)\right)}{\text{Tr} e^{-\int dy \sum_i \beta^i(y) q_i(y)}}. \tag{52}$$

We need to make one assumption about *homogeneous* GGE expectation values, which is that they satisfy the clustering properties,

$$\lim_{a_1,\ldots,a_{n-1}\to\infty} \frac{\text{Tr}\left(e^{-\sum_i \beta^i Q_i} \mathcal{O}_1(x,0)\mathcal{O}_2(x+a_1,0)\ldots\mathcal{O}_n(x+a_1+\cdots+a_{n-1},0)\right)}{\text{Tr} e^{-\sum_i \beta^i Q_i}}$$
$$= \frac{\text{Tr}\left(e^{-\sum_i \beta^i Q_i}\mathcal{O}_1\right)}{\text{Tr} e^{-\sum_i \beta^i Q_i}} \frac{\text{Tr}\left(e^{-\sum_i \beta^i Q_i}\mathcal{O}_2\right)}{\text{Tr} e^{-\sum_i \beta^i Q_i}} \times \cdots \times \frac{\text{Tr}\left(e^{-\sum_i \beta^i Q_i}\mathcal{O}_n\right)}{\text{Tr} e^{-\sum_i \beta^i Q_i}}, \tag{53}$$

that is, if local operators are very well separated in space, then the expectation value factorizes into the product of each expectation value.

We start by considering a "coarse grained" approach where we exploit the assumption of piece-wise slow variation of the chemical potentials $\beta^i(y)$. We assume the chemical potentials vary slowly enough (up to a finite number well spaced of sudden jumps) that we can approximate,

$$\int dy \sum_i \beta^i(y) q_i(y) \approx \sum_K \int_{K-\frac{l}{2}}^{K+\frac{l}{2}} dy \sum_i \beta^i_K q_i(y), \tag{54}$$

where we have divided space into discrete intervals of size $l$, labeled by the index, $K$. Our assumption of piece-wise smoothness is reflected in the fact that we assume that within each interval, the chemical potential is approximately a constant. That is, we assume the chemical potential depends only on the interval $K$, and not on the particular location inside each interval. We assume the intervals are large enough that $l$ is much larger than any internal scale (correlation length) of the model, yet $l \ll L$ and $L/l \sim L$.

Under these assumptions (for large enough $l$), we then find that the different intervals are asymptotically

uncorrelated, that is

$$\frac{\langle\rho_p|\left(\int_{K_1-\frac{l}{2}}^{K_1+\frac{l}{2}} dy \sum_i \beta_{K_1}^i\, q_i(y)\right)\left(\int_{K_2-\frac{l}{2}}^{K_2+\frac{l}{2}} dy \sum_i \beta_{K_2}^i\, q_i(y)\right)|\rho_p\rangle}{\langle\rho_p|\left(\int_{K_1-\frac{l}{2}}^{K_1+\frac{l}{2}} dy \sum_i \beta_{K_1}^i\, q_i(y)\right)|\rho_p\rangle\langle\rho_p|\left(\int_{K_2-\frac{l}{2}}^{K_2+\frac{l}{2}} dy \sum_i \beta_{K_2}^i\, q_i(y)\right)|\rho_p\rangle} \approx 1, \tag{55}$$

up to corrections that vanish at large $l$.

The denominator of (52) can then be written as

$$D \approx \text{Tr}\left(\prod_K e^{-\int_{K-\frac{l}{2}}^{K+\frac{l}{2}} dy \sum_i \beta_K^i q_i(y)}\right), \tag{56}$$

which is justified by assuming that for large $l$, the commutator, $\left[\int_{K-\frac{l}{2}}^{K+\frac{l}{2}} dy \sum_i \beta_K^i q_i(y), \int_{K'-\frac{l}{2}}^{K'+\frac{l}{2}} dy \sum_i \beta_{K'}^i q_i(y)\right]$, is small for $K \neq K'$. We are free to multiply each of these factors in the denominator by a factor of "1" as

$$D = \text{Tr}\left(\prod_K e^{-\int_{K-\frac{l}{2}}^{K+\frac{l}{2}} dy \sum_i \bar\beta^i q_i(y)}\, e^{\int_{K-\frac{l}{2}}^{K+\frac{l}{2}} dy \sum_i \bar\beta^i q_i(y)}\, e^{-\int_{K-\frac{l}{2}}^{K+\frac{l}{2}} dy \sum_i \beta_K^i q_i(y)}\right). \tag{57}$$

where we have introduce an arbitrary, *spatially homogeneous* set of chemical potentials, $\{\bar\beta_i\}$. At this point these chemical potentials are completely arbitrary and we can fix them later to any value that is convenient.

By the fact that operators defined in different $K$ cells commute with each other, we can pull all of the factors of $e^{-\int_{K-\frac{l}{2}}^{K+\frac{l}{2}} dy \sum_i \bar\beta^i q_i(x)}$ in front of the product, so we can write

$$D = \text{Tr}\left(e^{-\bar\beta^i Q_i} \prod_K O_K\right), \tag{58}$$

with,

$$O_K = e^{\int_{K-\frac{l}{2}}^{K+\frac{l}{2}} dy \sum_i \bar\beta^i q_i(x)}\, e^{-\int_{K-\frac{l}{2}}^{K+\frac{l}{2}} dy \sum_i \beta_K^i q_i(y)} \tag{59}$$

We have reformulated the denominator, $D$, as a standard *homogenous* GGE average of a product of operators with local support on the regions $(K - l/2, K + l/2)$. Under the assumption of piece-wise slow variation of the chemical potentials, $\beta^i(y)$, we can choose $l$ large enough such that We can apply the clustering property (53), then we we can express the denominator as[1]

$$\frac{D}{\text{Tr}\, e^{-\sum_i \bar\beta^i Q_i}} = \prod_K \frac{\text{Tr}\left(e^{-\sum_i \bar\beta^i Q_i} O_K\right)}{\text{Tr}\, e^{-\sum_i \bar\beta^i Q_i}} \tag{60}$$

We now turn our attention to the numerator of (52). We can again divide $x$ into cells of size $l$. The assumption of piecewise-smoothness of $\beta^i(y)$ means that the local operator $\mathcal{O}(x, 0)$ will only have a non-trivial correlation function with operators in the cell $K_x$, which contains the point $x$. That is,

$$\left[e^{\int_{K-\frac{l}{2}}^{K+\frac{l}{2}} dy \sum_i \bar\beta^i q_i(x)}\, e^{\int_{K-\frac{l}{2}}^{K+\frac{l}{2}} dy \sum_i \bar\beta^i q_i(x)}\, e^{-\int_{K-\frac{l}{2}}^{K+\frac{l}{2}} dy \sum_i \beta_K^i q_i(y)}, \mathcal{O}(x, 0)\right] \approx 0, \quad \text{for} \quad K \neq K_x, \tag{61}$$

---

[1]We point out that in this case we are assuming that the clustering property (53) is also applicable to semilocal operators, $O_K$, which have a support in a thermodynamically small region $l \ll L$. This can be justified for instance by further discretizing space with a lattice spacing $a = l/n$, with $n$ being a large integer, such that $\int_{K-\frac{l}{2}}^{K+\frac{l}{2}} dy \sum_i \beta_K^i q_i(y)$ becomes a discrete sum over $n$ completely local operators. The statement of the clustering property (63) is that the majority of these local operators within the semilocal operator $O_K$ are a large distance (of order $l$), from the operators in a different $O_{K'}$. Any corrections to the clustering property would come from local operators close to the edges $K \pm l/2$, however, as we take $l$ to be large, such operators close to the edge become a small minority, and their corrections to (60) would become subleading in powers of $l$.

where $K_x$ is defined such that $x \in (K_x - l/2, K_x + l/2)$. We can therefore write the numerator as

$$\mathcal{N} = \text{Tr}\left[e^{-\sum_i \bar{\beta}^i Q_i} \left(\prod_{K<K_x} O_K\right) O_{K_x} \mathcal{O}(x,0) \left(\prod_{K>K_x} O_K\right)\right]. \tag{62}$$

Again for a slowly enough varying set of chemical potentials, we can choose $l$ large enough such that we can apply the clustering properties on (62), such that

$$\langle \mathcal{O}(x,0)\rangle_{LGGE} = \frac{\mathcal{N}}{D} = \frac{\text{Tr}\left(e^{-\sum_i \bar{\beta}^i Q_i} O_{K_x} \mathcal{O}(x,0)\right)}{\text{Tr}\left(e^{-\sum_i \bar{\beta}^i Q_i} O_{K_x}\right)}, \tag{63}$$

where we have cancelled out factors containing $K \neq K_x$, between the numerator and denominator.

At this point we observe that we are still free to fix the set of parameters $\{\bar{\beta}^i\}$. The simplest choice we can make is to choose the chemical potentials such that $O_{K_x} = 1$. This can be done by choosing $\bar{\beta}^i = \beta^i_{K_x}$. In this case we have

$$\langle \mathcal{O}(x,0)\rangle_{LGGE} = \frac{\text{Tr}\left(e^{-\sum_i \beta^i_{K_x} Q_i} \mathcal{O}(x,0)\right)}{\text{Tr}\, e^{-\sum_i \beta^i_{K_x} Q_i}}. \tag{64}$$

We have thus shown the fact that at $t = 0$, local observables are described by their averages on a locally defined GGE, taking into account only the values of the chemical potential at the point $x$. The trace over states can also be replaced in the thermodynamic limit by a path integral over representative states, from which we can easily also express the expectation values as in Eq. (33).

In the next section we will see how a similar logic can be applied to the expectation value of local operators at late times, and we can derive the full GHD prediction (35)

## 7    Expectation values at late times: the Euler scale

We now study the expectation value

$$\langle \mathcal{O}(x,t)\rangle_{LGGE} = \frac{\text{Tr}\left(e^{-\int dy \sum_i \beta^i(y) q_i(y)} \mathcal{O}(x,t)\right)}{\text{Tr}\, e^{-\int dy \sum_i \beta^i(y) q_i(y)}} \tag{65}$$

for large values of $t$. We will do so by applying the same logic of dividing space into cells of size $l$. The denominator of (65) is the same as $D$ from last section in equation (58).

The main difference between the numerator of (65) and that of (52) from the previous section is that while the operator $\mathcal{O}(x,0)$ was only causally connected to the spatial cell labeled by $K_x$, the operator $\mathcal{O}(x,t)$ is causally connected to all the segments of the initial state which lie within the past lightcone of the operator. Following the same arguments as in the previous section, we can then write the numerator of (65) as

$$\mathcal{N} = \text{Tr}\left[e^{-\sum_i \bar{\beta}^i Q_i} \left(\prod_{K<K_-} O_K\right) \left(\prod_{K_-<K<K_x} O_K\right) O_{K_x} \mathcal{O}(x,t) \left(\prod_{K_x<K<K_+} O_K\right) \left(\prod_{K_+<K} O_K\right)\right] \tag{66}$$

where $O_K$ with $K \in (K_-, K_+)$ are all the operators in the initial state that lie within the past light-cone of $\mathcal{O}(x,t)$, such that $|(x - Kl)/t| < 1$, where the speed of light has been set to 1. We remark that again, we are still free to fix the values of $\bar{\beta}^i$ as is most convenient. Correlations between $\mathcal{O}(x,t)$ and operators outside of its lightcone are exponentially small, such that,

$$\frac{\mathcal{N}}{\text{Tr}\, e^{-\sum_i \bar{\beta}^i Q_i}} = \left(\frac{\text{Tr}\left[e^{-\sum_i \bar{\beta}^i Q_i} \left(\prod_{K_-<K<K_x} O_K\right) O_{K_x} \mathcal{O}(x,t) \left(\prod_{K_x<K<K_+} O_K\right)\right]}{\text{Tr}\, e^{-\sum_i \bar{\beta}^i Q_i}}\right)$$

$$\times \left(\frac{\text{Tr}\left[e^{-\sum_i \bar{\beta}^i Q_i} \left(\prod_{K<K_-} O_K\right)\right]}{\text{Tr}\, e^{-\sum_i \bar{\beta}^i Q_i}}\right) \left(\frac{\text{Tr}\left[e^{-\sum_i \bar{\beta}^i Q_i} \left(\prod_{K_+<K} O_K\right)\right]}{\text{Tr}\, e^{-\sum_i \bar{\beta}^i Q_i}}\right) + \mathcal{O}(e^{-\mu l}), \tag{67}$$

where any corrections are exponentially suppressed at large $l$.

With this factorization, we can now write the expectation value as

$$\langle \mathcal{O}(x,t)\rangle_{LGGE} = \frac{\mathcal{N}}{D} = \frac{\mathrm{Tr}\left[e^{-\bar{\beta}^i Q_i}\left(\prod_{K_- < K < K_x} O_K\right) O_{K_x}\mathcal{O}(x,t)\left(\prod_{K_x < K < K_+} O_K\right)\right]}{\mathrm{Tr}\left[e^{-\bar{\beta}^i Q_i}\left(\prod_{K_- < K < K_+} O_K\right)\right]}, \tag{68}$$

This expression can now be evaluated in terms of a form factor expansion between each pair of operators.

We can apply the quench action logic, and replace the trace over states by a path integral over particle density distributions, such that we arrive at

$$\langle \mathcal{O}(x,t)\rangle_{LGGE} = \frac{\mathcal{N}}{D} = \frac{\langle\rho_p^{\{\bar{\beta}\}}|\left(\prod_{K_- < K < K_x} O_K\right) O_{K_x}\mathcal{O}(x,t)\left(\prod_{K_x < K < K_+} O_K\right)|\rho_p^{\{\bar{\beta}\}}\rangle}{\langle\rho_p^{\{\bar{\beta}\}}|\left(\prod_{K_- < K < K_+} O_K\right)|\rho_p^{\{\bar{\beta}\}}\rangle}, \tag{69}$$

where the distribution $\rho_p^{\{\bar{\beta}\}}(\theta)$ is chosen to be the saddle point of the quench action,

$$S^{\{\bar{\beta}^i\}}[\rho_p] = \langle\rho_p|\sum_i \bar{\beta}^i Q_i|\rho_p\rangle + S_{YY}[\rho_p], \tag{70}$$

The expression (69) can now be evaluated by inserting intermediate sums over states between each pair of operators. At large times, contributions with a large number of particles and hole excitations on top of the representative state will be suppressed with higher powers of $1/t$. We then present the main proposal of this paper: *Generalized hydrodynamics arises from including only the leading contributions at asymptotically large $t$, involving zero-momentum only one-particle-hole pair form factors of the operators $O_K$ in the expression (69).*

We consider the first such contribution, arising from Inserting only up to one-particle-hole pair form factors in the (69), we arrive at the large-$t$ expansion[2]

$$\begin{aligned}\langle \mathcal{O}(x,t)\rangle_{LGGE} &= \langle\rho_p^{\{\bar{\beta}\}}|\mathcal{O}(0,0)|\rho_p^{\{\bar{\beta}\}}\rangle\\ &+\frac{1}{t}\sum_{K_- < K < K_+}\sum_{\theta\in\theta_*(\xi_K)}\lim_{\kappa_1,\kappa_2\to 0}\frac{n(\theta)(1-n(\theta))}{4\pi^2\rho_s(\theta)|(v^{\mathrm{eff}})'(\theta)|}\frac{f_{\rho_p^{\{\bar{\beta}\}}}^{O_K}(\theta+\pi i,\theta+\kappa_1)}{\langle\rho_p^{\{\bar{\beta}\}}|O_K|\rho_p^{\{\bar{\beta}\}}\rangle}\left(f_{\rho_p^{\{\bar{\beta}\}}}^{\mathcal{O}}(\theta+\pi i,\theta+\kappa_2)\right)^*\\ &+(\text{higher particle hole pairs contributions}),\end{aligned} \tag{72}$$

where we have defined

$$f_{\rho_p^{\{\bar{\beta}\}}}^{\mathcal{O}_K}(\theta_1,\theta_2) = \frac{\langle\rho_p^{\{\bar{\beta}\}}|O_K|\rho_p^{\{\bar{\beta}\}};\theta_1,\theta_2\rangle}{\langle\rho_p^{\{\bar{\beta}\}}|\rho_p^{\{\bar{\beta}\}}\rangle}, \tag{73}$$

and $\xi_K = (x - lK)/t$. We only keep in (69) terms which non-trivially correlate $\mathcal{O}$ other operators $O_K$. We do not keep similar terms which would correlate only $O_K$ and another $O_{K'}$ since, as we have argued, these operators lie outside of each other's lightcone, and such terms would decay exponentially.

We point out that even though the one-particle-hole pair form factor contribution to (72) looks naively like it decays linearly at late times, it actually has a leading contribution which survives at infinite times.

---

[2]To arrive at the expression 72 we have used repeatedly that form factors satisfy

$$\begin{aligned}\lim_{\kappa,\kappa'\to 0}\langle\theta,\theta+\pi i+\kappa;\rho_p|O_K|\rho_p;\theta',\theta'+\pi i+\kappa'\rangle &= \lim_{\kappa,\kappa'\to 0}\left[\langle\theta,\theta+\pi i+\kappa;\rho_p|\rho_p;\theta',\theta'+\pi i+\kappa'\rangle\langle\rho_p|O_K|\rho_p\rangle\right.\\ &\left.+\langle\rho_p|O_K|\rho_p;\theta',\theta'+\pi i+\kappa',\theta+\pi i,\theta+\kappa\rangle\right],\end{aligned} \tag{71}$$

which follows from crossing symmetry [35]. The first term in (71) when used repeatedly, leads to terms as described in (69), connecting $\mathcal{O}$ with every operator $O_K$, and the second term in (71) concerning two-particle-hole pair form factors is dropped, as it leads to subleading corrections as will be explained later.

This is because we are also summing over $K_- < K < K_+$, or equivalently integrating over all spatial points $y$ in the initial state which are within the past lightcone of the operator $\mathcal{O}(x,t)$. The size of this spatial interval also grow linearly with $t$, such that an additional factor of $\sim t$ is expected to arise from this integration.

Additional contributions arise from considering terms correlating the operator $\mathcal{O}(x,t)$ with two different operators in the initial state, $O_{K_1}$, and $O_{K_2}$ through one-particle-pair excitations, with contribution, which we denote as $\mathcal{O}^{1ph}_{\{\bar{\beta}\}K_1,K_2}$ being

$$
\begin{aligned}
\mathcal{O}^{1ph}_{\{\bar{\beta}\}K_1,K_2} =&\ \lim_{t\to\infty} \sum_{K_-<K_1<K_+} \sum_{K_-<K_2<K_+} \fint \left(\prod_{i=1}^{4} \frac{d\theta_i}{2\pi}\right) n(\theta_1)n(\theta_3)(1-n(\theta_2))(1-n(\theta_4)) \\
&\times \frac{f^{O_{K_1}}_{\rho_p^{\{\bar{\beta}\}}}(\theta_2+\pi\mathrm{i},\theta_1) f^{O_{K_2}}_{\rho_p^{\{\bar{\beta}\}}}(\theta_4+\pi\mathrm{i},\theta_3)}{\langle \rho_p^{\{\bar{\beta}\}}|O_{K_1}O_{K_2}|\rho_p^{\{\bar{\beta}\}}\rangle} \left(f^{\mathcal{O}}_{\rho_p^{\{\bar{\beta}\}}}(\theta_2+\pi\mathrm{i},\theta_4+\pi\mathrm{i},\theta_1,\theta_3)\right)^* \\
&\ \exp\{\mathrm{i}t\,[\xi_1(k(\theta_1)-k(\theta_2))+\xi_2(k(\theta_3)-k(\theta_4))-(\omega(\theta_1)+\omega(\theta_3)-\omega(\theta_2)-\omega(\theta_4))]\}. \quad (74)
\end{aligned}
$$

One can again perform a stationary phase approximation, where the leading contribution comes from terms proportional to $\frac{1}{t^2} f^{O_{K_1}}_{\rho_p^{\{\bar{\beta}\}}}(\theta_*(\xi_{K_1})+\pi\mathrm{i},\theta_*(\xi_{K_1})) f^{O_{K_2}}_{\rho_p^{\{\bar{\beta}\}}}(\theta_*(\xi_{K_2})+\pi\mathrm{i},\theta_*(\xi_{K_2}))$. Again, despite the explicit factor of $1/t^2$, there is a double integration over the spatial positions, $K_1, K_2$, so this yields a non-vanishing contribution at late times.

The full expectation value at late times then can be written as

$$
\langle \mathcal{O}(x,t)\rangle_{LGGE} \to \sum_{n=0} \sum_{\{K\}_n} C^{1ph}_{\{\bar{\beta}\}\{K\}_n}, \quad (75)
$$

where $\{K\}_n$ denotes a set of $n$ operators $O_{K_i}$, with $i=1,\ldots,n$ and the second sum is over their different possible values of $K_i$. The term $C^{1ph}_{\{\bar{\beta}\}\{K\}_n}$ contains terms proportional to $\frac{1}{t^n} f^{O_{K_1}}_{\rho_p^{\{\bar{\beta}\}}}(\theta_*(\xi_{K_1})+\pi\mathrm{i},\theta_*(\xi_{K_1}))\ldots f^{O_{K_n}}_{\rho_p^{\{\bar{\beta}\}}}(\theta_*(\xi_{K_n})+\pi\mathrm{i},\theta_*(\xi_{K_n}))$. After summing over all values of $K_i$, this again yields a leading order $t^0$ contribution. We have also defined $C^{1ph}_{\{\bar{\beta}\}\{K\}_0} \equiv \langle\rho_p^{\{\bar{\beta}\}}|\mathcal{O}|\rho_p^{\{\bar{\beta}\}}\rangle$ for consistency of notation.

We made it explicit in our notation that the terms $C^{1ph}_{\{\bar{\beta}\}\{K\}_n}$ depend on our choice of chemical potentials $\{\bar{\beta}\}$. In particular, we want to look for the choice of $\{\bar{\beta}\}_{\mathrm{sim}}$ which maximally simplifies the expressions for $C^{1ph}_{\{\bar{\beta}\}\{K\}_n}$ (the index "sim" stands for "simplest"). For this, it is important to point out that in integrable QFT's, knowledge of the complete set of local (and quasilocal) charges are sufficient to reconstruct the full particle distribution, or equivalently, the full filling fraction, $n(\theta)$ [49–53]. We therefore can talk about directly choosing a filling fraction associated with $|\rho_p^{\{\bar{\beta}\}}\rangle$, that simplifies the expression (72), without having to specify which set of chemical potentials reproduce this distribution. This means we can write

$$
\sum_i \bar{\beta}^i Q_i = \int d\theta' \mathbf{h}(\theta') A^\dagger(\theta') A(\theta'), \quad (76)
$$

where $\mathbf{h}(\theta) = \sum_i \bar{\beta}^i h_i(\theta)$, and $A^\dagger(\theta)$, $A(\theta)$ are particle creation and annihilation operators, respectively. That is, specifying the full set of chemical potentials $\{\bar{\beta}^i\}$, one for each conserved charge, is equivalent to specifying the function $\mathbf{h}(\theta)$, which can be understood as specifying a chemical potential corresponding to each value of rapidity, rather than each charge.

We now examine the form factors

$$
\begin{aligned}
&\lim_{\kappa\to 0} f^{O_K}_{\rho_p^{\{\bar{\beta}\}}}(\theta+\pi\mathrm{i},\theta+\kappa)|_{\theta=\theta_*(\xi_K)} \\
&= \lim_{\kappa\to 0} \frac{\langle \rho_p^{\{\bar{\beta}\}}| e^{\int_{K-\frac{l}{2}}^{K+\frac{l}{2}} dy \sum_i \bar{\beta}^i q_i(y)} e^{-\int_{K-\frac{l}{2}}^{K+\frac{l}{2}} dy \sum_i \beta^i_K q_i(y)}|\rho_p^{\{\bar{\beta}\}};\theta+\pi\mathrm{i},\theta+\kappa\rangle}{\langle \rho_p^{\{\bar{\beta}\}}|\rho_p^{\{\bar{\beta}\}}\rangle} \Bigg|_{\theta=\theta_*(\xi_K)}. \quad (77)
\end{aligned}
$$

In the previous section, in Eq (64), we made the choice $O_{K_x} = 1$, by choosing $\bar{\beta}^i = \beta^i_{K_x}$. The present case is not that simple, because now we have a large set of form factors, for different values of $K$, so we cannot choose a single set $\bar{\beta}^i$ that will make $O_K = 1$ for all $K \in (K_-, K_+)$. We can instead make a similar choice, but we only need to enforce a much weaker condition, since we only need the one-particle hole pair form factor at the particular rapidity value of $\theta_*(\xi_K)$ to vanish. Therefore we only need the projection of $O_K$ on a state with this value of rapidity to yield a vanishing form factor, instead of the operator completely vanishing as a whole, such that

$$\lim_{\kappa \to 0} f^{O_K}_{\rho_p^{\{\bar{\beta}\}}}(\theta + \pi\mathrm{i}, \theta + \kappa)|_{\theta = \theta_*(\xi_K)} \approx 0, \tag{78}$$

We proceed by expanding the exponentials in (77) as

$$
\begin{aligned}
&\lim_{\kappa \to 0} f^{O_K}_{\rho_p^{\{\bar{\beta}\}}}(\theta + \pi\mathrm{i}, \theta + \kappa)|_{\theta = \theta_*(\xi_K)} \\
&= \lim_{\kappa \to 0} \frac{\langle \rho_p^{\{\bar{\beta}\}}| \sum_{k,q=0}^{\infty} \left( \int_{K-\frac{l}{2}}^{K+\frac{l}{2}} dy \sum_i \bar{\beta}^i\, q_i(y) \right)^k \left( - \int_{K-\frac{l}{2}}^{K+\frac{l}{2}} dy \sum_i \beta^i_K q_i(y) \right)^q / (k! q!) |\rho_p^{\{\bar{\beta}\}}; \theta + \pi\mathrm{i}, \theta + \kappa\rangle}{\langle \rho_p^{\{\bar{\beta}\}}|\rho_p^{\{\bar{\beta}\}}\rangle} \Bigg|_{\theta = \theta_*(\xi_K)},
\end{aligned}
\tag{79}
$$

which we can evaluate term by term for different values of $k, q$.

The zeroth order term of (79) with $k = q = 0$ trivially vanishes as $\langle \rho_p^{\{\bar{\beta}\}}|\rho_p^{\{\bar{\beta}\}}; \theta + \pi\mathrm{i}, \theta + \kappa\rangle = 0$ by orthogonality. The next order includes terms such that $k + q = 1$, and is given by

$$\lim_{\kappa \to 0} \frac{\langle \rho_p^{\{\bar{\beta}\}}| \int_{K-\frac{l}{2}}^{K+\frac{l}{2}} dy \sum_i \bar{\beta}^i q_i(y) - \sum_i \beta^i_K\, q_i(y)|\rho_p^{\{\bar{\beta}\}}; \theta + \pi\mathrm{i}, \theta + \kappa\rangle}{\langle \rho_p^{\{\bar{\beta}\}}|\rho_p^{\{\bar{\beta}\}}\rangle} \Bigg|_{\theta = \theta_*(\xi_K)}. \tag{80}$$

We then want to chose the chemical potentials $\{\bar{\beta}^i\}$ such that (80) vanishes for any $K$. The condition on the function $\mathbf{h}(\theta)$ to acheive (80) at this order, for large $l$, is then given by

$$\mathbf{h}^{\mathrm{dr}}(\theta_*(\xi_K))\Big|_{\{\bar{\beta}\} = \{\bar{\beta}\}_{\mathrm{sim}}} = \left( \sum_i \beta^i_K h_i \right)^{\mathrm{dr}}(\theta_*(\xi_K)), \tag{81}$$

which has been derived by making use of the form factor of a conserved charge density (50) and dressing through the procedure described in (5) with a filling fraction $n(\theta)$ corresponding to our choice of $\{\bar{\beta}^i\}$ (equivalently choice of $\mathbf{h}(\theta)$).

It can also be shown that for large enough $l$, all the terms in the expansion (79) with higher values of $k, q$ also vanish upon making the choice (81). For these higher order terms, we generally have to compute the matrix element of a product of charge densities, $q_i(y)$, integrated over the cell $y = \left( K - \frac{l}{2}, K + \frac{l}{2} \right)$. We thus generally need to compute expressions of the form,

$$\lim_{\kappa \to 0} \frac{\langle \rho_p^{\{\bar{\beta}\}}| \left( \int_{K-\frac{l}{2}}^{K+\frac{l}{2}} dy\, q_i(y) \right)^n |\rho_p^{\{\bar{\beta}\}}; \theta + \pi\mathrm{i}, \theta + \kappa\rangle}{\langle \rho_p^{\{\bar{\beta}\}}|\rho_p^{\{\bar{\beta}\}}\rangle} \Bigg|_{\theta = \theta_*(\xi_K)}. \tag{82}$$

These can be computed in terms of a form factor expansion, by introducing a complete set of states between

any two of the charges,

$$
\lim_{\kappa \to 0} \frac{\langle \rho_p^{\{\bar{\beta}\}}| \left( \int_{K-\frac{l}{2}}^{K+\frac{l}{2}} dy\, q_i(y) \right)^n |\rho_p^{\{\bar{\beta}\}}; \theta + \pi\mathrm{i}, \theta + \kappa\rangle}{\langle \rho_p^{\{\bar{\beta}\}}|\rho_p^{\{\bar{\beta}\}}\rangle} \Bigg|_{\theta = \theta_*(\xi_K)}
$$

$$
= \sum_{\{\theta\}} \lim_{\kappa \to 0} \frac{\langle \rho_p^{\{\bar{\beta}\}}| \left( \int_{K-\frac{l}{2}}^{K+\frac{l}{2}} dy\, q_i(y) \right)^{n-1} |\rho_p^{\{\bar{\beta}\}}; \{\theta\}\rangle}{\langle \rho_p^{\{\bar{\beta}\}}|\rho_p^{\{\bar{\beta}\}}\rangle} \Bigg|_{\theta = \theta_*(\xi_K)}
$$

$$
\times \lim_{\kappa \to 0} \frac{\langle \rho_p^{\{\bar{\beta}\}}; \{\theta\}| \left( \int_{K-\frac{l}{2}}^{K+\frac{l}{2}} dy\, q_i(y) \right) |\rho_p^{\{\bar{\beta}\}}; \theta + \pi\mathrm{i}, \theta + \kappa\rangle}{\langle \rho_p^{\{\bar{\beta}\}}|\rho_p^{\{\bar{\beta}\}}\rangle} \Bigg|_{\theta = \theta_*(\xi_K)}. \tag{83}
$$

The integration over $y$ for large enough $l$ means that most intermediate states $\{\theta\}$ will give oscillatory contributions that vanish upon integration. The only surviving contributions are those where the set of excitations $\{\theta\}$ has zero momentum. That is, the only surviving contributions for large $l$ are those where the intermediate state contains only a set of zero-momentum particle-hole pairs on top of the representative state, or $\{\theta\} = \{\theta'\}, \{\theta' + \pi\mathrm{i}\}$. It then follows from the annihilation pole axiom derived in Eq. (1.10) of [28] that such form factors are proportional to the form factor (80), or

$$
\lim_{\kappa \to 0} \frac{\langle \rho_p^{\{\bar{\beta}\}}; \{\theta'\}, \{\theta' + \pi\mathrm{i}\}| \left( \int_{K-\frac{l}{2}}^{K+\frac{l}{2}} dy\, q_i(y) \right) |\rho_p^{\{\bar{\beta}\}}; \theta + \pi\mathrm{i}, \theta + \kappa\rangle}{\langle \rho_p^{\{\bar{\beta}\}}|\rho_p^{\{\bar{\beta}\}}\rangle} \Bigg|_{\theta = \theta_*(\xi_K)}
$$

$$
\sim \lim_{\kappa \to 0} \frac{\langle \rho_p^{\{\bar{\beta}\}}| \int_{K-\frac{l}{2}}^{K+\frac{l}{2}} dy\, q_i(y) |\rho_p^{\{\bar{\beta}\}}; \theta + \pi\mathrm{i}, \theta + \kappa\rangle}{\langle \rho_p^{\{\bar{\beta}\}}|\rho_p^{\{\bar{\beta}\}}\rangle} \Bigg|_{\theta = \theta_*(\xi_K)}. \tag{84}
$$

It therefore follows that if the condition (81) is satisfied, these form factors vanish, and thus the expression (83) vanishes. It is then easy to see that given the fact that (80) vanishes, after some rearranging of terms, all the higher terms in the expansion (79) fully cancel each other out. We have thus shown that by making the choice (81), the form factor $\lim_{\kappa \to 0} f_{\rho_p^{\{\bar{\beta}\}}}^{O_K}(\theta + \pi\mathrm{i}, \theta + \kappa)|_{\theta = \theta_*(\xi_K)}$ vanishes for all $K$.

It is now simple to see that the filling fraction $n(\theta)$ that satisfies Eq. (81) is exactly the one that satisfies the GHD evolution equation (36). For each value of rapidity $\theta$ the filling fraction is that which is specified by the values of chemical potentials, at the region, $K$ of the initial state, where a particle travelling at velocity $v^{\mathrm{eff}}(\theta)$ would reach the point $x$, where the operator is, at time $t$, that is, the filling fraction is described the solution (37).

After making this choice of chemical potentials $\{\bar{\beta}\}_{\mathrm{sim}}$, It is then evident that $C^{1ph}_{\{\bar{\beta}\}\{K\}_n} = 0$ for $n \neq 0$. the terms corresponding to one-particle-hole pair form factors in (72) vanish, leaving us with,

$$
\langle \mathcal{O}(x,t)\rangle_{LGGE} = C^{1ph}_{\{\bar{\beta}\}_{\mathrm{sim}}\{K\}_0} = \langle \rho_{p[x,t]}^{\mathrm{GHD}}|\mathcal{O}(0,0)|\rho_{p[x,t]}^{\mathrm{GHD}}\rangle
$$
$$
+ (\text{decaying terms}), \tag{85}
$$

where we now use the explicit notation, $|\rho_{p[x,t]}^{\mathrm{GHD}}\rangle$, to express that this is exactly the representative state which arises as the solution of the GHD time evolution (36).

We therefore have shown how generalized hydrodynamics emerges purely from the form factor expansion and quench action approach, as the leading term concerning only one-particle-hole pair form factors at zero

momentum. GHD then has the clear interpretation as the leading term in the form factor expansion, making it also clear what one needs to do to compute further corrections to the GHD, which is, to include higher form factors.

# 8 Leading quantum corrections from higher form factors

Now that we have understood what kind of approximation the GHD description of expectation value of local operators is, namely keeping only the zero-momentum one-particle-hole pair form factors contributions, it is easy to see what are the leading corrections. We divide these corrections into three broad categories: those corresponding to subleading corrections to the stationary phase approximation of the one-particle-hole pair form factor terms, terms involving a higher number of particle-hole pairs, and terms containing an unequal number of particles and holes.

The first kind of terms, concerning the one-particle-hole pair form factors, contain infomation about the non-zero momentum parts of the one-particle-hole pair form factor. A stationary phase approximation was used to arrive at the expression (51). Generally one needs to integrate over all rapidities of the particles and holes, however at late times the integrand of the one-particle-hole pair contribution becomes highly oscillatory, and the leading contribution to the stationary phase approximation keeps only the zero-momentum form factor, where the rapidities of the particle and the hole are equal. There are, however, computable corrections to this, which would come at order $1/t^2$ in expression (51). These would lead to order $1/t$ corrections to expression (72).

The second kind of corrections arise from considering terms including, for example, two-particle-hole-pair form factors for a particular operator $O_K$. The leading such contribution is

$$
\begin{aligned}
\lim_{t\to\infty} t\, \mathcal{O}^{2ph}_{\{\bar\beta\}K} &\equiv \lim_{t\to\infty} t \sum_{K_-<K_1<K_+} \fint \left( \prod_{i=1}^4 \frac{d\theta_i}{2\pi} \right) n(\theta_1)n(\theta_3)(1-n(\theta_2))(1-n(\theta_4)) \\
&\times f^{O_K}_{\rho_p^{\{\bar\beta\}}}(\theta_2+\pi\mathrm{i},\theta_4+\pi\mathrm{i},\theta_1,\theta_3) \left( f^{\mathcal{O}}_{\rho_p^{\{\bar\beta\}}}(\theta_2+\pi\mathrm{i},\theta_4+\pi\mathrm{i},\theta_1,\theta_3) \right)^* \\
&\exp\{\mathrm{i}t\left[\xi(k(\theta_1)+k(\theta_3)-k(\theta_2)-k(\theta_4))-(\omega(\theta_1)+\omega(\theta_3)-\omega(\theta_2)-\omega(\theta_4))\right]\}. \quad (86)
\end{aligned}
$$

One can again perform a stationary phase approximation, where the integration over rapidities gives a leading contribution of order $1/t^2$. Here, we again recover a factor of $t$ by summing over all $K$, however, in this case this is not enough to compensate for the factor of $1/t^2$. Therefore we have $\mathcal{O}^{2ph}_{\{\bar\beta\}K} \sim 1/t$. One can similarly consider other terms involving more than one operator, $O_K$, and one or more of them with two or more corresponding particle hole form factors, which would contribute at order $1/t$ or higher.

Lastly, we discuss contributions from terms with unequal numbers of particles and holes. The first such contribution comes from one-particle (or one hole) form factors, yielding a contribution,

$$
\mathcal{O}^{1p}_{\{\beta\}K} = \sum_{K_-<K<K_+} \sum \frac{d\theta}{2\pi}(1-n(\theta))f^{O_K}_{\rho_p^{\{\bar\beta\}}}(\theta)\left( f^{\mathcal{O}}_{\rho_p^{\{\bar\beta\}}}(\theta) \right)^* \exp\{\mathrm{i}t\left[\xi k(\theta)-\omega(\theta)\right]\} \quad (87)
$$

Again at large times, we can consider the stationary phase approximation, giving the leading contribution

$$
\mathcal{O}^{1p}_{\{\beta\}K} \sim \sum_{K_-<K<K_+} \sum_{\theta\in\theta_*(\xi_K)} \frac{1}{\sqrt{t}}(1-n(\theta))\sqrt{\frac{1}{\rho_s(\theta)(v^{\mathrm{eff}})'(\theta)}} f^{O_K}_{\rho_p^{\{\bar\beta\}}}(\theta)\left( f^{\mathcal{O}}_{\rho_p^{\{\bar\beta\}}}(\theta) \right)^* \exp\{\mathrm{i}t\left[\xi k(\theta)-\omega(\theta)\right]\}. \quad (88)
$$

This naively decays as $t^{-1/2}$, however we expect the decay to be further suppressed, by the fact the at the factor $\exp\{\mathrm{i}t\left[\xi k(\theta)-\omega(\theta)\right]\}$ is highly oscillatory. When integrating over values of $K$, this can again be done by an additional stationary phase approximation, yielding an additional factor of $t^{-1/2}$. The one-particle contribution should then decay as $t^{-1}$.

We note that we considered the inhomogeneous quench problem where the initial state is already very smooth, so that we can already divide the initial state into uncorrelated cells. Additional corrections to our formalism will arise of higher orders in $l/L$, when we consider less smooth initial states. It is possible that these may contribute with different powers of $t$ than the terms we have considered here.

While it is easy to understand where the leading corrections to the GHD limit come from, the physical intuition to obtain from these leading corrections is not that simple. Corrections within the GHD formalism have been explored by different means [26, 27, 54], where it is generally expected that diffusive effects are introduced. Seeing if and how these different extensions of GHD are recovered from our next-to-leading order form factor corrections, is however, we think a separate rich and interesting problem that should be explored in a future publication. For now we conform ourselves with recovering the standard GHD results from the form factor expansion.

# 9 Conclusions

We have shown how GHD time-evolution of local observables can be understood in terms of a form factor expansion. For a certain class of spatially inhomogeneous initial states, we have shown GHD descriptions arises from considering only up to the zero-momentum one-particle-hole pair form factors within the quench action formalism. These form factors are defined by excitations on top of a thermodynamic representative state. We can compute exactly this leading expression in relativistic QFT using recent results from the thermodynamic bootstrap program.

After understanding this limit, it is easy to see how one can add quantum corrections, simply by including more contributions from higher form factors.

It would be a very interesting question in the future to see if and how these higher form factor corrections can reproduce recent extensions of the GHD formalism, such as [26, 27, 54], or if these expansions coincide. Our expansion is organized in terms of powers of $1/t$, which are related to the number of particles and holes included in the form factors. It may be that the expansions proposed in [26, 27, 54], correspond to some reorganization of our subleading terms, where some partial resummation may be needed to show equivalence of the two methods.

Our results concern the expectation values of one-point functions after a spatially-inhomogeneous quench. It would also be interesting to try to extend these results to the case of higher point functions. There exist predictions for such $n$-point functions on inhomogenous states within the GHD formalism [8], so it would be interesting to see if the form factor expansion can recover these expressions as well.

## Acknowledgments

I thank Miłosz Panfil for a careful reading of the manuscript and many helpful suggestions, and Jean-Sébastien Caux for many fruitful discussions. This research received support from the European Research Council under ERC Advanced grant 743032 DYNAMINT.

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
