# Peer review of "How generalized hydrodynamics time evolution arises from a form factor expansion"

_SciPost Physics_

## Round 1 · Referee Report · Anonymous (Referee 2) · 2021-4-10

Strengths

1- Interesting and timely topic. 2- Systematic approach.

Weaknesses

1- Many assumptions are hard to test. 2- The discussion of corrections is incomplete.

Report

I apologise for the delay in submitting this report. Due to some technical issues with the journal website I learnt only very recently about new versions of this paper being uploaded. This report follows my first one submitted on the 23rd of April 2020.

I think that the author responded satisfactorily to most of my comments. However, before recommending the paper for publication, I strongly encourage the author to address the remaining points below.

Requested changes

1- I find the new paragraph added at the end of Section 5 rather misleading. The fact that one can reduce the form factor expansion to the form in Eq. 51 in the Euler limit is strictly speaking an assumption. Indeed, to get this result one has to assume that form factors corresponding to more particle-hole excitations are well behaved and one can perform the saddle point analysis mentioned by the author. I understand that this assumption was made earlier in the literature (in Ref. [29]) but I think that it would be better to explain this point also in the current paper. I also suggest the author to mention that the calculation presented in the paper is based on a number of non-trivial assumptions both in the introduction and in the conclusions.

2- The author should read carefully the paper again, removing the typos and spurious sentences introduced during revision. For example I found typos/spurious sentences in the discussion above Eq. 42, in the sentence above Eq. 47, and in the sentence above Eq 72. Finally, I am not sure about the grammar of the last sentence of the new paragraph in Sec. 5.

---

## Round 1 · Author Response

I thank the referees again for their thoughtful reports. I have fixed the minor issues that were raised by the referee, (mainly some typos and stylistic changes) so I hope the manuscript is now ready to be accepted for publication.

---

## Editorial Decision

awaiting_resubmission